# Incorporation of Poly(Ionic Liquid) with PVDF-HFP-Based Polymer Electrolyte for All-Solid-State Lithium-Ion Batteries

**DOI:** 10.3390/polym14101950

**Published:** 2022-05-11

**Authors:** Zhefei Ruan, Yuzhe Du, Hongfei Pan, Ruiming Zhang, Fangfang Zhang, Haolin Tang, Haining Zhang

**Affiliations:** 1State Key Laboratory of Advanced Technology for Materials Synthesis and Processing, Wuhan University of Technology, Wuhan 430070, China; ruanzhefei@foxmail.com (Z.R.); duyuzhe.whut@foxmail.com (Y.D.); thln@whut.edu.cn (H.T.); 2Foshan Xianhu Laboratory of the Advanced Energy Science and Technology Guangdong Laboratory, Foshan 528200, China; panhongfei@whut.edu.cn; 3Guangdong Hydrogen Energy Institute of WHUT, Foshan 528000, China; zhang_ruiming@yahoo.com; 4School of Nuclear Technology and Chemistry & Biology, Hubei University of Science and Technology, Nr. 88 Xianning Avenue, Xianning 437100, China

**Keywords:** solid polymer electrolyte, all-solid-state, lithium-ion battery, ion conduction, polymer composite

## Abstract

A solid-state polymer electrolyte membrane is formed by blending poly(vinylidene fluoride-co-hexafluoropropylene) with the synthesized copolymer of poly(methyl methacrylate-co-1-vinyl-3-butyl-imidazolium bis(trifluoromethanesulfonyl)imide, in which lithium bis(trifluoromethane)sulfonimide molecules are applied as the source of lithium ions. The accordingly formed membrane that contains 14 wt.% of P(MMA-co-VBIm-TFSI), 56 wt.% of PVDF-HFP, and 30 wt.% of LiTFSI manifests the best electrochemical properties, achieving an ionic conductivity of 1.11 × 10^−4^ S·cm^−1^ at 30 °C and 4.26 × 10^−4^ S·cm^−1^ at 80 °C, a Li-ion transference number of 0.36, and a wide electrochemical stability window of 4.7 V (vs. Li/Li^+^). The thus-assembled all-solid-state lithium-ion battery of LiFePO_4_/SPE/Li delivers a discharge specific capacity of 148 mAh·g^−1^ in the initial charge–discharge cycle at 0.1 C under 60 °C. The capacity retention of the cell is 95.2% after 50 cycles at 0.1 C and the Coulombic efficiency remains close to 100% during the cycling process.

## 1. Introduction

After three decades of development, the lithium-ion battery (LIB) has become the most popular secondary battery because of its high energy density, high operating voltage, low self-discharging rate and good cycling stability. However, there are still safety issues arising from the flammable organic liquid electrolytes contained in the LIB. In recent years, solid-state electrolytes (SSEs) have become the focus of attention due to their excellent properties, including nonflammability, favorable thermal stability, wide electrochemical window, and enhanced compatibility with Li metal anodes [1,2]. SSEs can be classified into three major categories: inorganic ion conductors, solid polymer electrolytes, and organic–inorganic hybrid composites [3]. Compared to inorganic ion conductors, the advantages of solid polymer electrolytes (SPEs) include small interfacial resistance, good flexibility, and low cost [4]. The relatively low ionic conductivity of polymer electrolytes remains problematic. Thus, improving the ionic conductivity of solid polymer electrolytes is of great importance for their practical application.

The ionic transport mechanism of SPEs has been controversial. The widely-accepted theory is that Li ions coordinate with the polar functional groups in the polymer chains, hopping from one interaction site to another, which mostly occurs in the amorphous parts of the polymer electrolyte [5]. The segmental relaxation and lithium-ion conduction of the SPEs are coupled phenomena [6]. Therefore, the ionic transport in SPEs is mainly associated with segmental motions. As a result, the ionic conductivity of SPEs can be promoted by decreasing the crystallinity and improving the flexibility of polymer chains. 

Poly(methyl methacrylate) (PMMA) and poly(vinylidene difluoride) (PVDF) are two commonly used polymers for SPEs. The carbonyl groups in the PMMA matrix provide coordination sites for Li ions, which makes lithium salts easier to dissociate. It also exhibits great dielectric properties and enhanced interfacial stability, but the poor mechanical flexibility hinders its practical application [7]. PVDF manifests high polarity and has a large dielectric constant owing to the strong electron-withdrawing group C–F, which assists in the dissolution of lithium salts [8]. Besides, it exhibits excellent thermal stability, mechanical strength, and chemical inertness [9]. However, PVDF shows a relatively high degree of crystallinity under room temperature (40–70%) [10]. To address the shortcomings of these two polymers, a variety of strategies have been suggested, such as blending with other polymers, incorporating other monomers into their matrixes, and introducing inorganic fillers to the systems. PMMA is usually used as a blender due to its good compatibility with other polymers [11]. Polymer blending combines the synergistic advantages to offset the respective weaknesses of different polymers [12]. For instance, blending PMMA with PEO decreases the crystallinity of PEO and the brittleness of PMMA [13]. Copolymer PVDF-HFP has been demonstrated to be a promising polymer host, whose structure consists of a crystalline region (formed by VDF unit) and an amorphous region (formed by HFP unit) [14]. The crystalline region promotes tensile strength and chemical stability, while the amorphous region reduces chain regularity and crystallinity [15]. The addition of nanoscale inorganic fillers such as Al_2_O_3_ or SiO_2_ can also enhance the tensile strength, interfacial stability, and ionic conductivity of SPEs by disrupting polymer crystallinity [16,17,18,19].

Ionic liquids (ILs), which are salts with a low melting point less than 100 °C, are considered solvents of the future because of their nonflammability, nonvolatility, high ionic conductivity, and wide electrochemical stability window. ILs have been applied to the electrolyte system of LIBs and have attracted tremendous attention in the past decade. They can be used as electrolyte solvents [20,21], additives in solid-state electrolytes [22,23], or fillers in polymer frameworks to obtain gel polymer electrolytes [24,25,26,27]. When ionic liquid units are linked together with themselves or other monomers, poly(ionic liquid)s (PILs) are formed, reflecting a combination of the excellent properties of both ILs and polymers [28,29,30].

In this study, methyl methacrylate was copolymerized with imidazolium-based ILs in order to produce a copolymer with high flexibility and ionic conductivity. As the most studied IL cation, imidazolium has a weak binding effect with anion, and is easy to graft with different functional groups. As a result, imidazolium-based ILs are characterized by low viscosity, high ionic conductivity and flexibility in design [31]. Bis(trifluoromethanesulfonyl)imide (TFSI^−^), compared with other IL anions, shows better thermal and electrochemical stability [32,33,34,35]. Hence, the IL 1-vinyl-3-butyl-imidazolium bromide (VBIm-Br) was chosen to be the polymeric monomer. After copolymerization, bromide ions are replaced by TFSI^−^ via the anion exchange process. The thus-synthesized copolymer was blended with PVDF-HFP for further improvement of the electrochemical properties. The lithium salt, lithium bis(trifluoromethanesulfonyl)imide (LiTFSI) was added to the system as the lithium-ion donor.

## 2. Materials and Methods

### 2.1. Material

1-Vinylimidazole (99%), 2, 2′-Azobis(2-methylpropionitrile) (AIBN) (99%, recrystallized), N, N-Dimethylformamide (DMF) (99.5%), methyl methacralyte (MMA), and lithium bis(trifluoromethanesulfonyl)imide (LiTFSI) were purchased from Aladdin. Ethyl acetate (99%), acetonitrile (99%), N-butyl bromide (98%), and poly(vinylidene fluoride-co-hexafluoropropylene) (PVDF-HFP) (average Mw~455,000) were purchased from Rhawn. Nano-scale LiFePO_4_ (99%) was purchased from Shenzhen Dynanonic. All other reagents were used as received.

### 2.2. Synthesis of the Ionic Liquid Monomer VBIm-Br

VBIm-Br was synthesized by using a traditional quaternary ammonization method according to the literature [36]. 1-Vinylimidazole and N-butyl bromide (mole ratio 1:1) were dissolved in ethyl acetate, refluxed at 80 °C for 20 h under nitrogen atmosphere. After static stratification of the solution, the hot ethyl acetate on the upper layer was removed, and the cool ethyl acetate was added. The sticky ionic liquid VBIm-Br in the lower layer was intensely stirred to make it recrystallized into white grains, which were subsequently vacuum-dried at 40 °C for 18 h to remove residue solvent.

### 2.3. Synthesis of the Copolymer P(MMA-co-VBIm-TFSI)

P(MMA-co-VBIm-Br) was synthesized by using a traditional free radical polymerization process. MMA and VBIm-Br (mole ratio 7:3) were dissolved in acetonitrile using AIBN (1 wt.% with respect to the monomers) as the initiator, refluxed at 65 °C for 16 h under nitrogen atmosphere. The solution was then added to a large amount of deionized water, allowing the raw product to precipitate as flocculent deposit, which was subsequently washed and dried using a lypholizer. Finally, P(MMA-co-VBIm-TFSI) was synthesized by anion exchange reaction. P(MMA-co-VBIm-Br) and LiTFSI (mole ratio 2:3) were dissolved in methanol separately, and then the P(MMA-co-VBIm-Br) solution was slowly added to the LiTFSI solution. After stirring for 12 h, the mixture was added to a large amount of deionized water to make P(MMA-co-VBIm-TFSI) precipitate to form a powdery deposit, which was subsequently washed and vacuum-dried at 70 °C for 12 h.

### 2.4. Preparation of Polymer Electrolyte Membranes

The polymer electrolyte membranes with different weight compositions were prepared by the solution casting method. PVDF-HFP, P(MMA-co-VBIm-TFSI), and LiTFSI were dissolved in DMF, then stirred for 12 h to obtain a homogeneous solution. The mixture was cast into a Teflon tray, vacuum-dried at 60 °C for 20 h, and then at 80 °C for 4 h. Finally, the membranes were cut into discs and stored in an argon-filled glove box. Lithium salts work as a Li-ion donor as well as a plasticizer in membranes. The addition of lithium salts increases the concentration of free Li-ions, but reduces the mechanical strength of the membranes. The optimized LiTFSI concentration was found to be 30%. The composition of the polymer electrolyte membranes are showed in Table 1.

### 2.5. Assembly of Cells

The cathode was composed of 80 wt.% LiFePO_4_, 10 wt.% acetylene black as the conductive agent, and 10 wt.% PVDF as the binder. The materials were dissolved in *N*-methyl-pyrrolidone (NMP), then stirred for 12 h to obtain a homogeneous paste. The paste was cast on aluminum foil and vacuum-dried at 70 °C for 12 h, and then air-blast-dried at 100 °C for 4 h. CR2016 button cells were assembled by sandwiching the SPE membrane between the LiFePO_4_ cathode and the Li metal anode in an argon-filled glove box.

### 2.6. Characterization

The functional groups and chemical bonds of the SPEs were confirmed by Fourier transform infrared spectroscopy (Thermo Scientific Nicolet 6700, Thermo Scientific, Waltham, MA, USA) over the wavenumber range of 4000 cm^−1^ to 400 cm^−1^. X-ray diffraction (Bruker D8 Advance, Bruker, Billerica, MA, USA) with Cu-Kα radiation was applied to analyze the crystallinity of the SPEs with different compositions. The diffraction angle (2θ) was set between 5° and 70° with a scan rate of 10° per minute. The melting point, glass transition temperature, and thermal stability of the SPEs were investigated by simultaneous thermal analysis (Netzsch STA449F3, Netzsch, Selb, Germany). Thermogravimetric (TG) and the differential scanning calorimetry (DSC) were conducted from 30 to 700 °C and −150 to 120 °C, respectively, with a heating rate of 10 °C per minute under nitrogen atmosphere. The mechanical properties of the SPEs were characterized by an electrical tension tester (Instron 5967, Instron, Norwood, MA, USA) at room temperature. The surface morphology of the SPEs was investigated by scanning electron microscopy (JEOL JSM-7500F, JEOL, Tokyo, Japan).

### 2.7. Electrochemical Properties

Electrochemical measurements were carried out on a multifunctional electrochemical workstation (Princeton VersaSTAT3, Princeton, Dublin, Ireland). The ionic conductivity (σ) of the SPE was measured by electrochemical impedance spectroscopy (EIS) over the frequency range of 100 kHz to 1 Hz under an AC amplitude of 10 mV at various temperatures (from 30 °C to 80 °C, measured every 10 °C) with the SS (stainless steel)/SPE/SS cell, calculated using the following equation:(1)$$\sigma =\frac{L}{R_{b}\cdot S}$$
where *R_b_* is the bulk electrolyte resistance, and *L* and *S* are the thickness and contact area of the electrolyte membrane, respectively [37].

The electrochemical stability window of the SPE was evaluated by linear sweep voltammetry (LSV) and cyclic voltammetry (CV) at 60 °C with the SS/SPE/Li cell. LSV curves were recorded from 0 to 6 V with a scan rate of 10 mV·s^−1^, and CV curves were measured from −1 to 1.5 V with a scan rate of 0.5 mV·s^−1^.

The interfacial resistance between the lithium electrode and SPE was evaluated by EIS, and the Li-ion transference number of the SPEs was measured by combining EIS and DC polarization at 60 °C with the Li/SPE/Li asymmetric cell, calculated by the Bruce–Vincent equation:(2)$$t_{Li^{+}}=\frac{I_{s}\left(\Delta V-I_{0}R_{0}\right)}{I_{0}\left(\Delta V-I_{s}R_{s}\right)}$$
where *I*_0_ and *I_S_* are the initial and stable state polarization current, *R*_0_ and *R_S_* are the initial and stable state interfacial resistance, respectively, Δ*V* is the applied polarization voltage. EIS was recorded over the frequency range of 100 kHz to 1 Hz under an AC amplitude of 10 mV. DC polarization was carried for 3000 s under an applied voltage of 10mV.

Furthermore, the cycling performance of the LiFePO_4_/SPE/Li full cell was evaluated by the galvanostatic charge–discharge tests from 2.5 V to 3.8 V at 60 °C on the battery cycler system (LAND CT2001A, LAND, Wuhan, China).

## 3. Results and Discussion

The designed poly(ionic liquid)s were synthesized through traditional free radical polymerization using AIBN as the initiator in the presence of MMA and pre-synthesized VBIm-Br, as schematically shown in Figure 1. FTIR spectra of P(MMA-co-VBIm-Br) and P(MMA-co-VBIm-TFSI) are shown in Figure 2a to qualitatively investigate the formation of poly(ionic liquid)s. The characteristic absorption bands at 1729 cm^−1^ and 1150 cm^−1^ originate from the stretching vibration of C=O and C-O-C bonds of MMA. The absorption band at 1569 cm^−1^ corresponds to the skeletal vibration of the imidazole ring. Moreover, the weak absorption band at 1633 cm^−1^ attributed to the stretching vibration of C=C suggests that most of the monomers has been polymerized. This indicates the successful synthesis of P(MMA-co-VBIm-Br). The synthesized copolymer is a random type according to the literature due to the similar reactivity ratio of the two applied monomers [36]. In addition, three new peaks appear in the spectra of P(MMA-co-VBIm-TFSI) after replacing bromide anions with TFSI^−^ through ion exchange process. The absorption bands at 1352 cm^−1^ and 1193 cm^−1^ originate from the asymmetric and symmetric stretching vibration of S=O bond of TFSI^−^, respectively. The absorption band at 1058 cm^−1^ corresponds to the stretching vibration of C-F bond of TFSI^−^, demonstrating that the anion exchange reaction has been accomplished. 

The strain–stress curves of the thus-cast SPE membranes with different compositions are displayed in Figure 2b. It is apparent that the addition of P(MMA-co-VBIm-TFSI) has a negative effect on the mechanical properties, resulting in the decline in both the tensile strength and breaking elongation rate. The introduction of ILs into PMMA chains improves the flexibility and weakens the mechanical strength. As can be seen, a membrane with a weight ratio of P(MMA-co-VBIm-TFSI) to PVDF-HFP that exceeds 3:7 becomes too fragile to be practically applied in batteries. Thus, membrane consisting of 20 wt.% poly(ionic liquid) moieties were applied in the following electrochemical measurements to ensure high ionic conductivity and reasonable mechanical strength. The XRD patterns of SPE membranes are shown in Figure 2c. With the increase in the content of P(MMA-co-VBIm-TFSI), a decrease in the intensity of the diffraction peak at around 2θ of 20° is observed. In addition, the intensity of two diffraction peaks at 2θ of around 37° and 39° decreased with the addition of P(MMA-co-VBIm-TFSI), which nearly disappear in the pattern of the PIL-30% sample. It can be thus concluded that the addition of P(MMA-co-VBIm-TFSI) to PVDF-HFP reduces the crystallinity of the SPE by augmenting the amorphous region, promoting the ionic conductivity of the SPE.

SEM images of the fabricated SPE membranes are showed in Figure 3. It can be observed that the blended membranes present a coarse and porous surface structure, and the surface of the PVDF-HFP/LiTFSI membrane is relatively smooth. SEM images with higher magnification are shown in the insets in Figure 3. Apparently, the crevices on the membrane become larger with the addition of P(MMA-co-VBIm-TFSI). This phenomenon suggests that the addition of P(MMA-co-VBIm-TFSI) to PVDF-HFP leads to a decrease in surface density, thus weakening the mechanical strength of the SPE membranes.

TG and DSC curves of the fabricated SPE membranes are presented in Figure 4. As observed in Figure 4a, a slight weight loss occurs from room temperature to about 150 °C due to the moisture evaporation. A massive weight loss starts at around 300 °C, resulting from the decomposition of the lithium salt and polymer backbone. With the addition of P(MMA-co-VBIm-TFSI), the decomposition temperature of the SPE slightly increases owing to the highly stable TFSI^−^ anion, suggesting that the thermal stability can be improved by blending the IL copolymer with conventional polymer electrolyte. From the DSC curves in Figure 4b, it can be seen that the melting temperature (T_m_) and glass transition temperature (T_g_) of the SPE decrease with the addition of P(MMA-co-VBIm-TFSI). The decrease in T_m_ further demonstrates the reduced crystallinity, and the decrease in T_g_ suggests the increase in chain flexibility, both of which facilitate the mobility of the SPE chain segments, thus improving the ionic conductivity of the SPE.

Figure 5a shows the temperature-dependent ionic conductivity of SPE membranes. It can be observed that the ionic conductivity increases initially with the addition of P(MMA-co-VBIm-TFSI), reaching a peak value at the PIL content of 20%. With a further increase in the content of P(MMA-co-VBIm-TFSI), the ionic conductivity of the membrane starts to decrease. This phenomenon can be understood in that the addition of PIL fraction reduces the crystallinity and improves the flexibility of the SPE, thus accelerating the segmental motion of the polymer chains which in turn enhances the lithium-ion migration. Membranes containing either PIL or PVDF-HFP were prepared and the temperature-dependent ionic conductivity is displayed in Figure 5b. Since lithium salts can act as plasticizer in SPE membranes, the content of lithium salt is reduced from 30% to 15% to make the formation of PIL membrane with certain mechanical strength successful. Because the bulk ionic conductivity of P(MMA-co-VBIm-TFSI)/LiTFSI is less than PVDF-HFP/LiTFSI, the total ionic conductivity of the SPE decreases if excess PIL is added. Thus, the optimized weight ratio of PIL to PVDF-HFP is 1:4 with which the SPE exhibits a maximum ionic conductivity of 1.11 × 10^−4^ S·cm^−1^ and 4.26 × 10^−4^ S·cm^−1^ at 30 °C and 80 °C, respectively. It is found that temperature dependence of ionic conductivity of pure PVDF-HFP or PIL SPE is almost linear, thus suggests that its ion transport mechanism is Arrhenius type, implying that the ion transport occurs via a simple hopping mechanism decoupled from the segmental motion of polymer chains. However, temperature-dependent ionic conductivity of the blended modified SPE is non-linear, suggesting that its ionic transport mechanism is Vogel–Tamman–Fulcher (VTF) type, which implies that its ion transport involves polymer segmental relaxation [38]. These phenomena further indicate that improving the chain flexibility and accelerating the segmental motion are essential for enhancing ionic conductivity of the blended modified SPE.

The LSV curves of the membranes are displayed in Figure 6a to investigate the electrochemical windows of the fabricated SPEs. It is found that the electrochemical stability window of the PVDF-HFP membrane is approximately 4.7 V (vs. Li/Li^+^), whereas it slightly decreases with the addition of P(MMA-co-VBIm-TFSI). For example, an obvious drop from 4.7 to 4.6 V (vs. Li/Li^+^) is observed when the weight ratio of PIL to PVDF-HFP is up to 3:7. Similar phenomenon has been reported in previous research, which could be a result of the relatively low decomposition potential of PMMA (4.7 V vs. Li/Li^+^) in comparison to PVDF (5.0 V vs. Li/Li^+^) [39]. The PIL-20% sample exhibits the maximum ionic conductivity and an electrochemical stability window suitable for practical applications (about 4.7 V vs. Li/Li^+^) and the CV curve of such SPE is shown in Figure 6b. Only a lithium dissolution peak at 0.23 V (vs. Li/Li^+^) and lithium deposition peak at −0.27 V (vs. Li/Li^+^) were observed, further demonstrating its promising electrochemical stability.

The AC impedance spectra of the Li/SPE/Li symmetric cells assembled with the PIL-20% sample and PVDF-HFP sample after various storage times is displayed in Figure 6c. For comparison, the result of the PVDF-HFP sample is presented as the inset in the same figure and the derived values are displayed in Figure 6d. The initial interfacial resistances of the two samples are 75 and 76 Ω, very close to each other. Within a week the interfacial resistance of the PVDF-HFP sample increased from 75 to 109 Ω, whereas it increased from 76 to 96 Ω for the PIL-20% sample due to the formation of solid electrolyte interphase (SEI) between the SPE and Li metal. The lower resistance of the PIL-20% sample after a week of storage compared to the PVDF-HFP sample could be the result of the enhanced interfacial stability of PMMA, and the interaction between the IL and Li metal surface. It has been reported that IL with TFSI^−^ anion favors uniform lithium deposition on the surface of Li anode, forming a SEI layer with low resistance [40]. Therefore, the interfacial resistance between the SPE and Li metal can be reduced by the addition of P(MMA-co-VBIm-TFSI).

Current–time curves and AC impedance spectra of the Li/SPE/Li symmetric cells assembled with the PVDF-HFP sample and PIL-20% sample are displayed in Figure 7. Calculated by Bruce–Vincent equation, the Li-ion transference number of the PIL-20% sample is 0.36, much larger than that of the PVDF-HFP sample (0.21). This indicates that the migration of Li ions can be facilitated by the blending of P(MMA-co-VBIm-TFSI) with PVDF-HFP, due to the decreased crystallinity and improved segmental flexibility. At a high current rate, Li-ion transference number of electrolytes is a decisive factor in the cell performance. Therefore, the blending modification can increase the discharging capacity and reduce the capacity loss of the cell testing at high current rates.

Since the PIL-20% sample exhibits the best electrochemical properties and acceptable mechanical strength, it was thus applied as solid electrolyte for the assembly of an all-solid-state LiFePO_4_/SPE/Li battery. The electrochemical performance of the battery was tested from 2.5 to 3.8 V at 60 °C. For comparison, the cells assembled from the PVDF-HFP sample were also tested under the same condition. The initial charge–discharge capacity of the cells at 0.1 C is shown in Figure 8a. It can be observed that the cell assembled with the PIL-20% sample exhibits a discharge capacity of 146.0 mAh·g^−1^ with the coulombic efficiency of 99.9%, remarkably higher than those of the cell assembled from the PVDF-HFP (133.5 mAh·g^−1^ with the coulombic efficiency of 97.6%). The cycling performance of the cells at 0.1 C is displayed in Figure 8b. After 50 charge–discharge cycles, the capacity retention of the cell assembled from the PIL-20% sample is 95.2% with an average coulombic efficiency of about 98.0% whereas the cell assembled from the PVDF-HFP sample has a capacity retention of 97.5% with an average coulombic efficiency of about 98%. The high discharge capacity of the cell assembled with the PIL-20% sample is attributed to the high ionic conductivity and low interfacial resistance of the blended SPE membrane. The constant coulombic efficiency indicates that the stable SEI is formed between the SPE and electrodes.

The charge–discharge capacity and cycling performance at various current rates were further investigated, as presented in Figure 9. It can be observed that the discharge capacity of the cell assembled from the PVDF-HFP sample evidently decreases from 133.8 to 46.8 mAh·g^−1^ with the increase of the current rate from 0.1 to 1 C, implying a capacity loss of 65% at 1 C. After a further decrease in the current rate to 0.1 C, the discharge capacity of the assembled cell is about 122.8 mAh·g^−1^ with a capacity recovery of 91.8%. By comparison, the cell assembled from the PIL-20% sample shows a slight decrease in discharge capacity from 148.4 to 127.5 mAh·g^−1^ with the increase in the current rate from 0.1 to 1 C, and only 14.1% of the capacity loss at 1 C. Moreover, the capacity recovery is 95.4% (141.5 mAh·g^−1^) after the current rate is switched back to 0.1 C. The superior performance at a high current rate of the cell assembled with the PIL-20% sample is attributed to the excellent electrochemical properties, particularly the enhanced Li-ion transference number. This is because enhancing the Li-ion transference number reduces the anion accumulation on the anode interface, resulting in a decrease in the internal polarization resistance of the cell. These results clearly indicate that the blended modified SPE exerts a positive influence on the cell performance.

## 4. Conclusions

In summary, a novel SPE membrane was prepared by blending copolymer P(MMA-co-VBIm-TFSI) with PVDF-HFP. The addition of P(MMA-co-VBIm-TFSI) promotes the ionic conductivity and Li-ion transference number, reducing the interfacial resistance between the SPE and Li metal. However, the addition of poly(ionic liquid) fractions had a negative effect on the mechanical properties of the formed membranes, resulting in a decreased mechanical strength and a slightly decreased electrochemical stability window. Under an optimized composition, the resulting SPE exhibits the maximum ionic conductivity (1.11 × 10^−4^ S·cm^−1^ at 30 °C and 4.26 × 10^−4^ S·cm^−1^ at 80 °C), an electrochemical stability window suitable for application in lithium-ion cells (about 4.7 V vs. Li/Li^+^), acceptable mechanical properties and a promising Li-ion transference number (0.36). The accordingly assembled all-solid-state LiFePO_4_/SPE/Li cell shows a remarkable cycling performance under various current rates. After 50 cycles at 0.1 C under 60 °C, the capacity retention is 95.2% with an average coulombic efficiency of about 98%. A relatively high discharge capacity is obtained at 1 C (127.5 mAh·g^−1^) and the capacity recovery is 95.4% after the rate is switched back to 0.1 C.

## Figures and Tables

**Figure 1 polymers-14-01950-f001:**
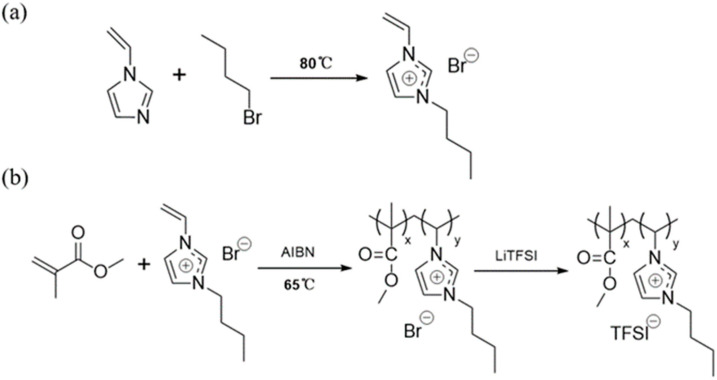
Synthesis of ionic liquid monomer VBIm-Br (**a**) and copolymer P(MMA-co-VBIm-TFSI) (**b**).

**Figure 2 polymers-14-01950-f002:**
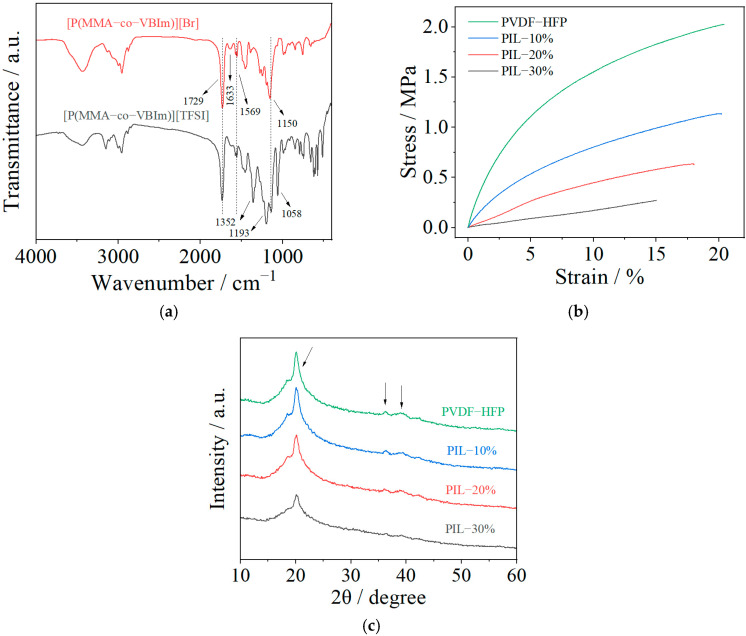
(**a**) FTIR spectra of P(MMA-co-VBIm-Br) and P(MMA-co-VBIm-TFSI). (**b**) Stress–strain curves and (**c**) XRD patterns of SPE membranes with different compositions as indicated in the figure.

**Figure 3 polymers-14-01950-f003:**
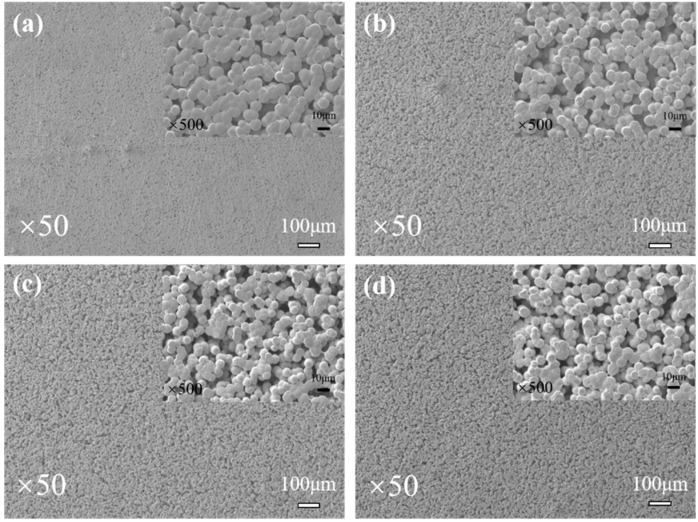
SEM images of SPE membranes with different compositions: (**a**) PVDF-HFP; (**b**) PIL-10%; (**c**) PIL-20%; (**d**) PIL-30%.

**Figure 4 polymers-14-01950-f004:**
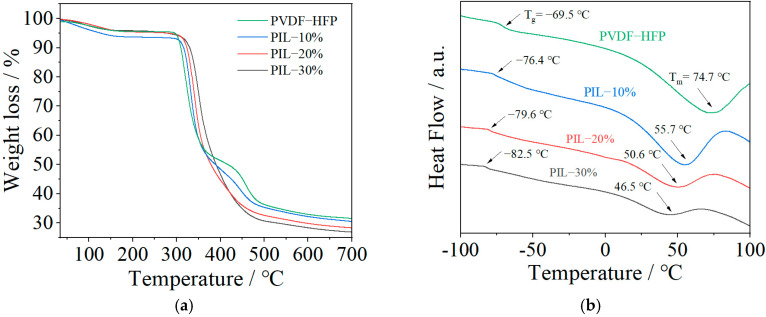
TG (**a**) and DSC (**b**) curves of the fabricated SPE membranes with different compositions.

**Figure 5 polymers-14-01950-f005:**
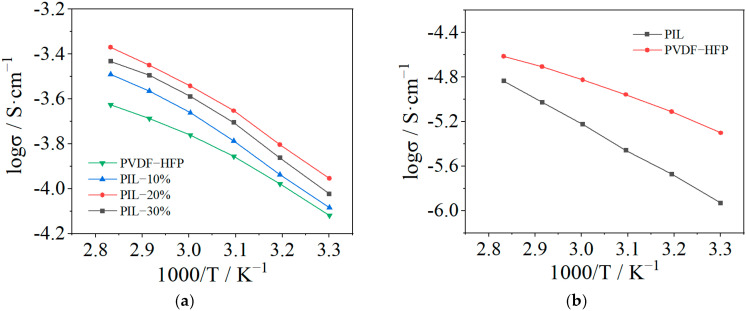
Temperature dependence of ionic conductivity of the SPE membranes with different weight ratio of PIL to PVDF-HFP (30 wt.% LiTFSI) (**a**) and containing either PIL or PVDF-HFP (15 wt.% LiTFSI) (**b**).

**Figure 6 polymers-14-01950-f006:**
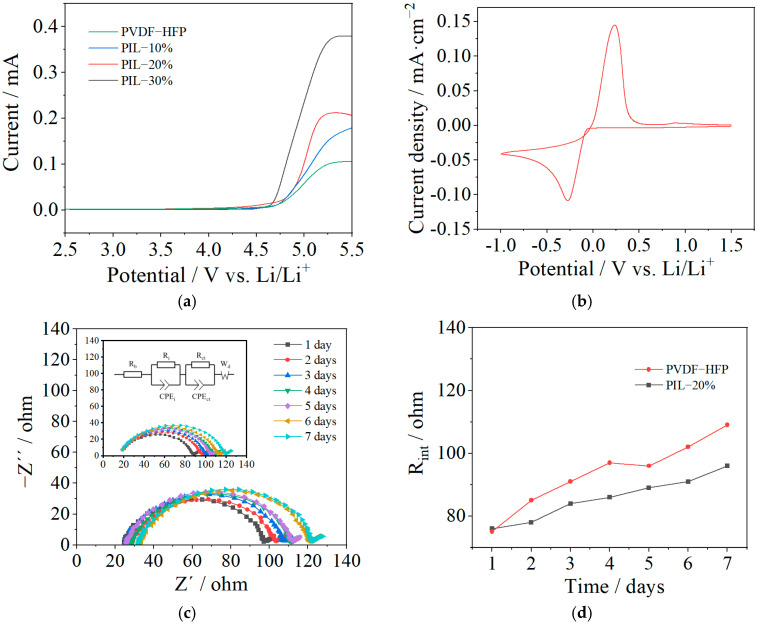
(**a**) LSV curves of the fabricated SPE membranes. (**b**) CV curve of the PIL-20% sample. (**c**) AC impedance spectra of the Li/SPE/Li symmetric cells assembled with the PIL-20% sample and PVDF-HFP sample after various storage times. (**d**) The interfacial resistance derived from AC impedance spectra.

**Figure 7 polymers-14-01950-f007:**
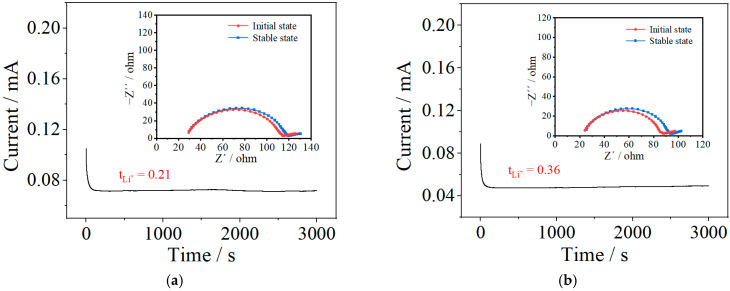
Current–time curves and AC impedance spectra (presented as the inset) of the Li/SPE/Li symmetric cells assembled with PVDF-HFP sample (**a**) and PIL-20% sample (**b**).

**Figure 8 polymers-14-01950-f008:**
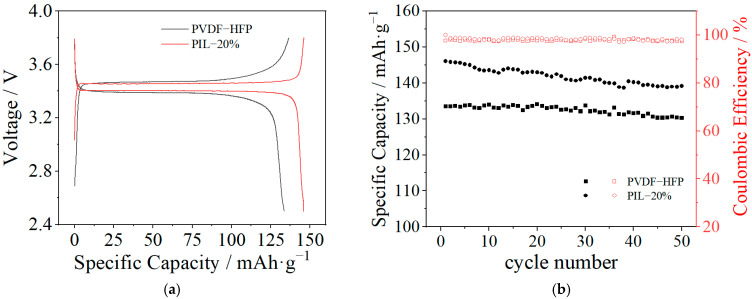
The initial charge–discharge capacity (**a**) and the cycling performance (**b**) of the cells assembled with the PVDF-HFP sample and PIL-20% sample at 0.1 C under 60 °C.

**Figure 9 polymers-14-01950-f009:**
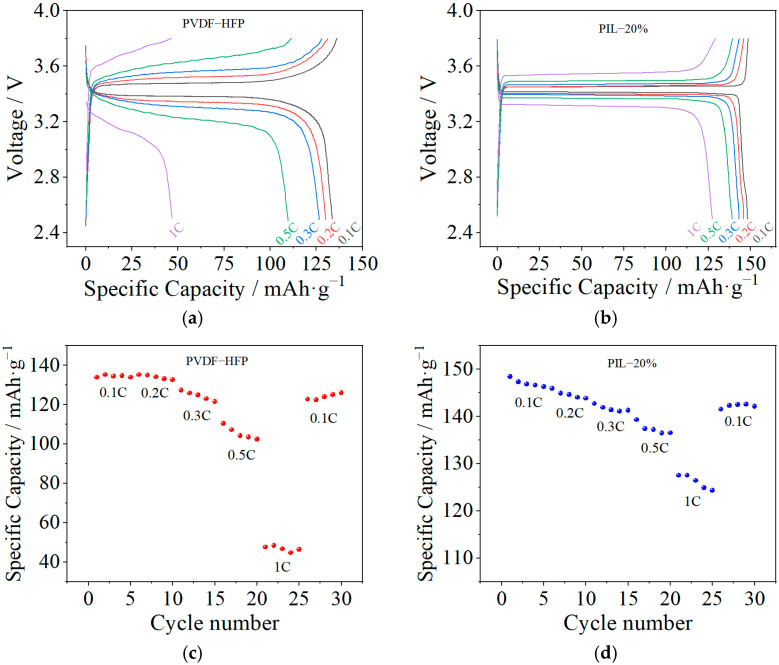
The charge–discharge capacity (**a**,**b**) and cycling performance (**c**,**d**) of cells assembled with PVDF-HFP and PIL-20% samples at various current rates under 60 °C.

**Table 1 polymers-14-01950-t001:** Composition of the prepared SPE membranes.

Sample	PMMA-IL (wt.%)	PVDF-HFP (wt.%)	LiTFSI (wt.%)	Weight Ratio of PMMA-IL/PVDF-HFP
PVDF-HFP	0	70	30	-
PIL-10%	7	63	30	1:9
PIL-20%	14	56	30	2:8
PIL-30%	21	49	30	3:7

## Data Availability

The data that support the findings of this study are available on request from the corresponding author H. Zhang.

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
