# Peer review of "Incorporation of Poly(Ionic Liquid) with PVDF-HFP-Based Polymer Electrolyte for All-Solid-State Lithium-Ion Batteries"

_polymers, 2022, doi:10.3390/polym14101950_

Round 1

Reviewer 1 Report

This article reports incorporation of poly(ionic liquid) with PVDF-HFP based polymer electrolyte for all-solid-state lithium ion batteries. A solid-state polymer electrolyte was formed by blending poly(vinylidene fluoride-co-hexafluoropropylene) with the synthesized random copolymer of poly(methyl methacrylate-co-1-vinyl-3-butyl-imidazolium bis(trifluoromethanesulfonyl)imide, in which lithium bis(trifluoromethane)sulfonimide was dissolved as the source of lithium ions. The work is interesting but major revision of the article is required before its suitability for publication in Polymers in the light of following comments:

  1. SPEs stand for what? Fist write complete name and then use abbreviataion.
  2. Correct the poorly written sentences in the introduction. e.g However, safety issue resulted from the applied flammable organic liquid electrolyte is the remained challenge during the practical applications.
  3. You did not prove that by using the described preparation procedure you obtained a random copolymer of poly(methyl methacrylate-co-1-vinyl-3-butyl-imidazolium bis(trifluoromethanesulfonyl)imide. Presenting a copolymer it is necessary to spectroscopically prove the presence of both types of mers in one polymer chain. Moreover, it is necessary to unequivocally determine which type of copolymer is formed: random?,alternating? block?
  4. Capacity retention of the cell achieves 95.2% after 50 cycles at 0.1 C. It is suggested to incorporate results for at least 100 cycles.
  5. Conclusions should be modified in the light of above comments.

Reviewer 2 Report

Provide description of the abbreviation SPE.

Sentence: “However, PVDF shows a relatively high degree of crystallinity under room temperature (40-60 %).“ Actually, crystallinity of PVDF can be higher. I recommend to increase values (40-70%) and cite for this reference: [doi: 10.3390/polym13152439]

I suggest to fitt XRD data and remove a part of spectra that does not provide information.

SEM images with heigher magnification should be inserted. 

More detailed description (text) of electrical behaviour should be added, because now it looks like only a report.

Reviewer 3 Report

The work «Incorporation of poly(ionic liquid) with PVDF-HFP based polymer electrolyte for all-solid-state lithium ion batteries» is devoted very interesting and actual topic. Developments and study of new electrode materials, electrolytes and their combinations to improve LIB characteristics and decrease their cost are important and wide-spreading part of world research. This manuscript has a large enough introduction to describe the research problem and the proposed method of solving it. All different methods used are described in detail. Figures shown are informative and good quality. Many experimental results are obtained and discussed. They have practical significant. But I have some comments:

  1. It is necessary to explain how the synthesis modes (time and temperature of drying) for "Synthesis of the ionic liquid monomer VBIm-Br" were chosen. Links to sources or otherwise.
  2. Why the authors used 30 wt.% LiTFSI concentration. In the text of the article, you must add a justification.
  3. For semi-cells, the authors used a cathode based on LiFePO4. However, there is no information about its origin, i.e. was it synthesized or acquired? In any case, it is necessary to give more information about its properties and method of obtaining. What justified the choice of this particular cathode material?
  4. Electrochemical testing was done at a temperature of 60 C. It is necessary to justify the choice of this temperature. At the same time, the DSC results show a change in the melting temperature (Tm) and glass transition temperature (Tg) from 45 to 75 C.
  5. Figures 5 a-b are Arrhenius plots, but there is no information about the temperatures at which the data were obtained. The main thing is that there is no analysis of the Arrhenius equation, there are no activation energy values, this equation is not in the "methods". Otherwise, why exactly this axes on graphics 5 a-b?
  6. To interpret impedance measurements, equivalent circuits and their analysis must be added. On Nyquist plots, the scale of the "X" and "Y" axes must be the same!

Round 2

Reviewer 1 Report

Though the authors have improved the revised version to certain level there still exist some issues to be addressed. For example one cannot determine the nature of copolymer (i.e. random, alternating or block) solely from FTIR spectroscopy. Similarly the cyclic stability test should be done for more cycles rather than confining the study to 50 cycles.

Author Response

We sincerely thank the reviewer for the effort made on this work. We do agree with the reviewer that FTIR cannot determine the random nature of the copolymer. We tried to explain the nature of the copolymer by using 13C NMR spectrscopy but failed possibly due to the weak intensity of the according carbon resonance. We thus deleted the word of "random" in the revised manuscript. 

We agree with the reviewer that capacity retention measurement after 50 cycles is not enough to characterize the durability of cells. In this work, we just like to introduce the concept that poly(ionic liquid)-based solid electrolyte might be possible for all-solid-state lithium ion batteries with reasonable modification and also due to the time limitation the long cyclic tests were not provided.

Reviewer 3 Report

Accept in present form.

Author Response

We sincerely thank the reviewer for the valuable comments. We have carefully checked the manuscript and all the typos and grammatia errors have been corrected as highlighted with red colors in the revised manuscript.